# Bias-aware Boolean Matrix Factorization Using Disentangled Representation Learning

**Xiao Wang**[1]  **Jia Wang**[1]  **Tong Zhao**[3]  **Yijie Wang**[1]  **Nan Zhang**[4,5]  **Yong Zang**[2]  **Sha Cao**[2]  **Chi Zhang**[2]

[1]Department of Computer Science, Indiana University, Bloomington, Indiana, USA
[2]School of Medicine, Indiana University, Indianapolis, Indiana, USA
[3]Uber Inc, Seattle, Washington, USA
[4]Institute of Science and Technology for Brain-inspired Intelligence, Fudan University, Shanghai, China
[5]School of Data Science, Fudan University, Shanghai, China

## Abstract

Boolean matrix factorization (BMF) has been widely utilized in fields such as recommendation systems, graph learning, text mining, and -omics data analysis. Traditional BMF methods decompose a binary matrix into the Boolean product of two lower-rank Boolean matrices plus homoscedastic random errors. However, real-world binary data typically involves biases arising from heterogeneous row- and column-wise signal distributions. Such biases can lead to suboptimal fitting and unexplainable predictions if not accounted for. In this study, we reconceptualize the binary data generation as the Boolean sum of three components: a binary pattern matrix, a background bias matrix influenced by heterogeneous row or column distributions, and random flipping errors. We introduce a novel Disentangled Representation Learning for Binary matrices (DRLB) method, which employs a dual auto-encoder network to reveal the true patterns. DRLB can be seamlessly integrated with existing BMF techniques to facilitate bias-aware BMF. Our experiments with both synthetic and real-world datasets show that DRLB significantly enhances the precision of traditional BMF methods while offering high scalability. Moreover, the bias matrix detected by DRLB accurately reflects the inherent biases in synthetic data, and the patterns identified in the bias-corrected real-world data exhibit enhanced interpretability.

## 1 INTRODUCTION

Boolean matrix factorization (BMF) seeks to detect lower-dimensional patterns in a binary matrix, which contrasts with traditional matrix factorization techniques that focus on real-valued matrices. BMF has various applications across different domains [Miettinen and Neumann, 2020]. One notable application is in recommender systems, where it can be used to extract meaningful user-item preferences or item-item similarities from binary user-item interaction data [Rukat et al., 2017, Balasubramaniam et al., 2018]. It can also be employed in bioinformatics for gene expression analysis [Liang et al., 2020, Zhang et al., 2007, 2010], network analysis [Kocayusufoglu et al., 2018a,b], and binary pattern mining Lucchese et al. [2010a,b, 2013]. Preserving the binary nature of the data poses unique challenges to BMF [Stockmeyer, 1975, Miettinen et al., 2008]. Several approaches have been proposed to address this issue, including non-negative matrix factorization (NMF) with binary constraints [Araujo et al., 2016], BMF with specific optimization objectives [Wan et al., 2020b, Lucchese et al., 2010a,b, 2013, Miettinen et al., 2008, Miettinen and Vreeken, 2011, 2014], and probabilistic models that capture the binary nature of the data [Ravanbakhsh et al., 2016, Rukat et al., 2017].

Despite the significant advancements in BMF, it is essential to recognize that the current BMF formulation, which regards a matrix as the sum of a series of low-rank Boolean pattern matrices and independent and identical (i.i.d) random errors [Ravanbakhsh et al., 2016, Rukat et al., 2017], overlooks the presence of biases within the data. When biases pervade the underlying data, the patterns extracted are likely to assimilate these biases, leading to a distortion of the resultant patterns [Mehrabi et al., 2021, Yao and Huang, 2017]. Such distortion can adversely affect the interpretation of the data, as well as subsequent analyses and decisions.

Real-world data often display unique biases and heteroscedastic (non-i.i.d.) distributions of signals and errors, which traditional BMF methods may not fully account for [Wan et al., 2020a]. A common form of bias arises from the varied distributions of row-/column-wise signals. For instance, in purchase history data, "super items" that are disproportionately popular can be purchased by a large number of users. Similarly, "super users", who purchase a wider array of items more frequently, can introduce further imbalances. These biases, driven by varying propensities, can

distort the representation of true underlying patterns. Traditional BMF approaches, which typically assume equal importance across all items or users, may struggle to differentiate between genuine patterns and the biases introduced by the prevalence of these super items or users.

To tackle the bias issue in BMF, it is imperative to incorporate bias-aware assumptions in solving BMF. Here, we present a novel approach where we rethink the generation of the binary matrix as the Boolean sum of the true and to-be-detected low-rank patterns, row-wise and column-wise biases, and heteroscedastic errors. Building upon this fundamental idea, we introduce **DRLB**[1] (**D**isentangled **R**epresentation **L**earning method of **B**inary matrix), a cutting-edge deep learning framework specifically designed to disentangle a Boolean matrix into two distinct components: a low-rank pattern matrix $(U \otimes V)$ and a bias matrix adept at capturing the row-wise and column-wise biases. The key contributions of this study include:

- DRLB reconceptualizes a Boolean matrix as the aggregation of patterns, background bias, and heteroscedastic random errors. This flexible model can be generally applied across diverse real-world datasets, expanding its utility and relevance.

- DRLB is the first deep neural network-based method to untangle the intricate relationship between low-rank patterns and the background bias inherent in rows and columns of a Boolean matrix.

- DRLB enhances the bias removal of Boolean matrices. It can be implemented with any existing BMF methods to facilitate more accurate data analyses and interpretations.

It is noteworthy that DRLB is developed to recognize and adjust for non-random and systematic biases in data, which is distinct from errors. DRLB specifically improves the awareness of skewness in Boolean data for a bias-aware BMF.

## 2 RELATED WORK

### 2.1 BOOLEAN MATRIX FACTORIZATION

For a given Boolean matrix $X \in \{0,1\}^{m \times n}$ of $m$ rows and $n$ columns, BMF computes a pair of low-rank binary matrices $(U \in \{0,1\}^{m \times k}$ and $V \in \{0,1\}^{k \times n})$, whose Boolean product (denoted as $\otimes$) approximates $X$, which could be generally presented as:

$$X \sim U \otimes V, \quad X_{ij} \sim \vee_{l=1}^{k} U_{il} \wedge V_{lj}. \quad (1)$$

Here, $\wedge$ and $\vee$ denote "and" and "or" operations. The two low-rank binary matrices are often solved under Boolean arithmetic to minimize the Frobenius norm or other norms of the reconstruction error $||E|| = ||X - U \otimes V||$. For example,

ASSO [Miettinen et al., 2008] builds a row-wise correlation matrix and employs a heuristic mechanism for retrieving binary base matrices, while PANDA Lucchese et al. [2010b] iteratively discovers and retains the most significant patterns. However, the high computational cost of these methods limits their application to large-scale datasets. MEBF [Wan et al., 2020b] utilizes binary matrix permutation theory and geometric segmentation that largely improved the efficiency in detecting 1s enriched patterns. A recent method, namely CG [Kovacs et al., 2020], significantly improved the pattern detection accuracy by formulating the BMF problem as a mixed integer linear program and introducing a column generation-based optimization. In recent years, multiple methods that follow the formulation of (1) have been developed to improve the accuracy and efficiency of BMF [Miettinen et al., 2008, Rukat et al., 2017, Kovacs et al., 2020, Lucchese et al., 2010b, Dalleiger and Vreeken, 2022, Avellaneda and Villemaire, 2022, Ravanbakhsh et al., 2016, Fischer and Vreeken, 2021, Neumann and Miettinen, 2020].

### 2.2 BIAS-AWARE BMF

Conventional Binary Matrix Factorization (BMF) algorithms have demonstrated commendable performance when the input follows the formulation of (1). However, this formulation may oversimplify the generation processes observed in real-world scenarios, in which data often exhibit a biased distribution of row-/column-wise signals, like the super users or items in purchase history data. Such biases will cause the conventional BMF methods to identify patterns that are skewed towards either the rows or columns displaying a high frequency of '1's, thereby impacting the accuracy and reliability of the subsequent pattern extraction and analysis. Specifically, certain patterns will be accentuated while others will be downplayed.

A recent study, BIND [Wan et al., 2020a], first introduced the consideration of row-/column-dependent background bias in BMF. BIND identifies and eliminates the entire rows and columns that are less likely to be contained by a low-rank pattern before BMF, thus improving detection accuracy and decreasing the reconstruction error in the presence of background bias. However, our analysis reveals that BIND may introduce additional bias (see EXPERIMENTS and Figure 2). BIND also assumes identical fitting errors, which further limits its applicability in real-world data analysis. Thus, it is crucial to develop robust bias-handling approaches that could comprehensively capture the generation process of Boolean matrices.

### 2.3 DISENTANGLED REPRESENTATION LEARNING

Disentangled representation learning aims to identify and disentangle the latent patterns in input data [Wang et al.,

---

[1]code is available at https://github.com/xwang97/DRLB

2022]. With the ability to decompose the observations into components carrying different types of information, disentangled learning has demonstrated its high interpretability of the input data in diverse applications. In computer vision and recommender systems, VAE and GAN-based models are exploited to disentangle independent factors of variation and manipulate latent variables [Higgins et al., 2016, Kim and Mnih, 2018, Chen et al., 2016, Ma et al., 2019a]. Cross-domain tasks could also be improved by embedding the input from different domains into a shared domain-invariant content space and disentangling this shared space from different domain-specific attribute spaces [Lee et al., 2018].

In this paper, we reconsider the generation of binary data as the Boolean sum of three components: patterns, background, and random error. This assumption can naturally adopt the advantage of disentangled representation learning. By training two auto-encoders to extract the pattern and background bias in an unsupervised fashion, we showed our model well captured the data generation process and achieved highly desirable performance in binary data bias removal.

# 3 DRLB FRAMEWORK

## 3.1 NOTATIONS

In this study, we represent a matrix, vector, and scalar value by uppercase ($X$), bold lowercase ($\mathbf{x}$), and lowercase ($x$) characters, respectively. The upper script represents the dimension of the object (e.g. $X^{m \times n}$), while the lower script indicates the element indices (e.g. $i$-th row: $X_{i:}$, $j$-th column: $X_{:j}$, and $ij$-th element: $X_{ij}$). $||\cdot||$ represents a general form of matrix norm, such as the Frobenius norm. Under Boolean arithmetic, the *and*, *or*, and *not* operations are denoted by $\wedge$, $\vee$, and $\neg$. Subsequently, the Boolean element-wise sum and subtraction are defined as $X \oplus Y = X \vee Y$ and $X \ominus Y = (\neg X \vee Y) \wedge (X \vee \neg Y)$. The Boolean matrix product is defined as $H = X \otimes Y$, where $H_{ij} = \vee_{l=1}^{k} X_{ik} \wedge Y_{lj}$.

## 3.2 MATHEMATICAL CONSIDERATIONS

We first consider the following generation approach of a Boolean matrix $X \in \{0,1\}^{m \times n}$:

$$X = U \otimes V \oplus X^B + E \qquad (2)$$

, where $X^B$ is the background bias matrix generated by row- and column-wise probability vectors $\mathbf{p}^r \in R^{m \times 1}$ and $\mathbf{p}^c \in R^{n \times 1}$, which reflect the probability of observing 1s in a row or a column of $X^B$:

$$P(X_{i:}^0 = 1) \propto \mathbf{p}^r \ and \ P(X_{:j}^0 = 1) \propto \mathbf{p}^c.$$

$E$ is an identical and independent flipping error with a flipping probability $P(1 \rightarrow 0) = P(0 \rightarrow 1) = p_0$. Noted, (2) extends (1) by introducing the background bias matrix $X^B$, which forms a bias-aware BMF problem. Intuitively,

(2) considers the observed 1s could be either generated by low-rank patterns or the row/column-dependent background bias. Thus, a successful disentanglement of $X^B$ may not only decrease the fitting error $|X \ominus X^B - U \otimes V|$ but also increase the explainability of the 1s in the observed data.

In real-world data, another challenge is that the distribution of error could be heterogeneous, i.e.,

$$P(1 \rightarrow 0) \neq P(0 \rightarrow 1).$$

Moreover, $P(1 \rightarrow 0)$ and $P(0 \rightarrow 1)$ may depend on $U$, $V$, $\mathbf{p}^r$, $\mathbf{p}^c$, or even $X^0$. For example, the density of 1's could vary among the sub-matrices, meaning $P(1 \rightarrow 0)$ depends on $U$ and $V$. To capture the generation of real-world Boolean data and enable robust BMF for more general inputs, we further extend (2) to the following probabilistic definition of the bias-aware BMF problem.

**Definition 1. Bias-Aware BMF (BABMF) problem.** For a given Boolean matrix $X \in \{0,1\}^{m \times n}$, BABMF seeks for the solution of $U^{m \times k}$, $V^{k \times n}$, $X^B$, such that $P(X_{i:}^B = 1) \propto \mathbf{p}^r$ and $P(X_{:j}^B = 1) \propto \mathbf{p}^c$, and

$$P(X_{ij} = 1) = \begin{cases} \mathbf{p}_i^c \mathbf{p}_j^r, & \text{if } (U \otimes V)_{ij} = 0 \\ min\{\mathbf{p}_i^c \mathbf{p}_j^r + p_{ij}, 1\}, & \text{if } (U \otimes V)_{ij} = 1 \end{cases}$$
$$(3)$$

, where $p_{ij}$ encodes the low-rank patterns different from bias or error, i.e., $p_{ij} > 0$ if $(U \otimes V)_{ij} = 1$, and $p_{ij} = 0$ otherwise. We want to note that (3) provides a general formulation of the bias-aware BMF problem. Without additional constraints on $p_{ij}$, (3) is incorrectly posed because $p_{ij}$ has a trivial solution under Maximum Likelihood Estimation (MLE). When the structure of $p_{ij}$ is given, an MLE-based solution could be formulated. Noted, (3) provides a rigorous probabilistic formulation of the generation approach of a Boolean matrix following (2).

In practice, it is always challenging to derive the analytic form of errors for real-world data, i.e., defining the structure of $p_{ij}$. Here we propose a heuristic and general solution to the BABMF problem formulated in (3) without any pre-assumption of errors, namely **D**isentangled **R**epresentation **L**earning based **B**MF (DRLB). DRLB considers that $X$ is formed by the Boolean sum of $X^B$ (background bias matrix) and $X^P$ (pattern matrix). As defined in (3), $X^P$ is generated by $P(X_{ij}^P = 1) = p_{ij}$ and $X^B$ is generated by $P(X_{ij}^B = 1) = \mathbf{p}_i^c \mathbf{p}_j^r$. DRLB disentangles the Boolean input $X$ into the Boolean sum of $X^P$ and $X^B$.

## 3.3 OVERVIEW OF THE NETWORK ARCHITECTURE OF DRLB

For a given Boolean matrix $X$, we aim to learn a latent representation $Z$ that could reconstruct the input through a neural network. The likelihood of $X$ can be represented by:

$$P(X) = \int_Z P(X|Z)P(Z)dZ \qquad (4)$$

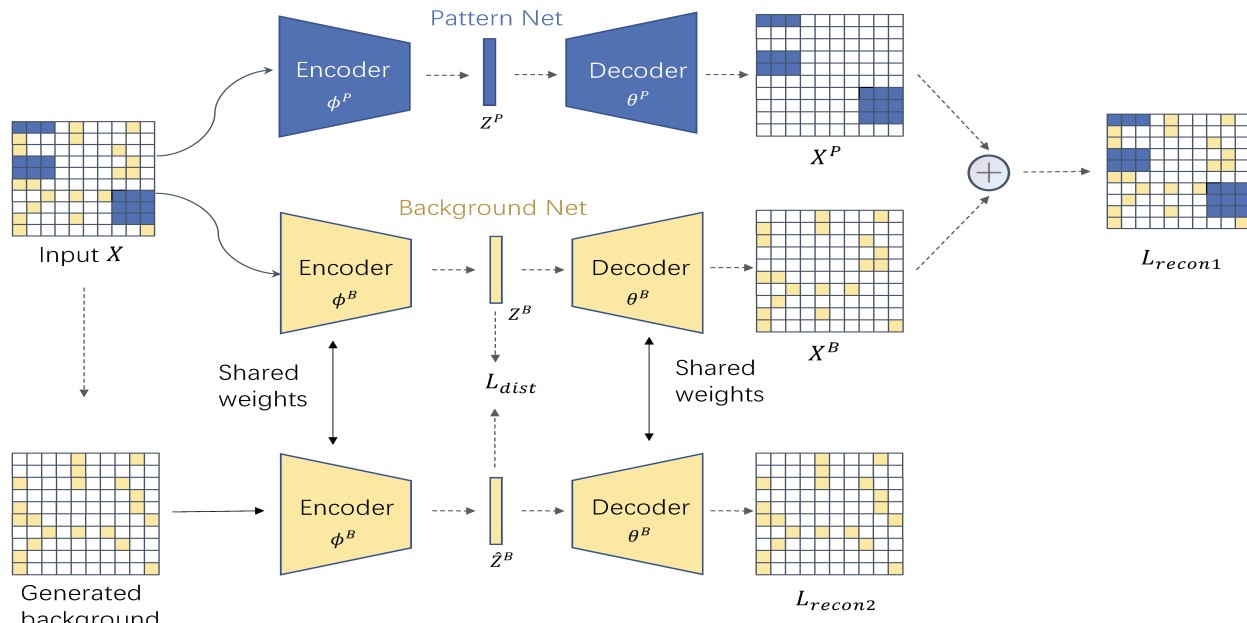

Figure 1: The DRLB Framework. The observed matrix serves as input for both the Pattern Net (blue) and Background Net (yellow). The outputs of the two decoders reconstruct the input by their Boolean sum. Background bias matrices are first generated based on the row-wise and column-wise distributions of the input matrix, which serves as the training input for the Background Net. The bottleneck features of the observed matrix and generated background extracted by the Background Net are constrained by a distribution loss. The Pattern Net and Background Net disentangle the input matrix into (1) the bias-removed pattern + error matrix and (2) background bias, which could be seamlessly input into a BMF method.

Previous studies on variational inference [Kingma and Welling, 2013] have derived that the log-likelihood has a lower bound as:

$$\log P(X) \geq \mathbf{E}_{Q_\phi(Z|X)}[\log P_\theta(X|Z)] - \mathbf{KL}(Q_\phi(Z|X)||P(Z))$$
(5)

, where $\phi$ presents the parameters of the neural network $Q_\phi(Z|X)$ (encoder) that approximates the posterior probability $P(Z|X)$, and $\theta$ represents the parameters of the neural network $P_\theta(X|Z)$ (decoder) of the likelihood $P(X|Z)$.

Following the definition of BABMF and the discussion in section 3.2, a binary matrix is generated by the Boolean sum of pattern and bias matrices, i.e., $X = X^P \oplus X^B$. In DRLB, we solve the BABMF problem by factorizing the latent representation of $Z$ into two independent components, $Z^P$ and $Z^B$, which separately generate the pattern matrix $X^P$ and bias matrix $X^B$. Under this consideration, (5) can be extended to (see detailed derivations in APPENDIX A.1):

$$\log P(X) \geq \mathbf{E}_{Q_{\phi^P, \phi^B}(Z^P, Z^B|X)}[\log P_{\theta^P, \theta^B}(X|Z^P, Z^B)]$$
$$-\mathbf{KL}(Q_{\phi^P}(Z^P|X)||P(Z^P)) - \mathbf{KL}(Q_{\phi^B}(Z^B|X)||P(Z^B))$$
(6)

To factorize $Z$, we introduced two neural networks (as illustrated in Figure 1), namely (1) Pattern Net (colored in blue, with parameters $\phi^P, \theta^P$) and (2) Background Net (colored

in yellow, with parameters $\phi^B, \theta^B$). In 3.4, we detail the collaborative training of the two networks and how the lower bound derived in (6) ensures an effective factorization of $Z$ and the disentanglement of the patterns and bias matrices.

## 3.4 DISENTANGLED REPRESENTATION LEARNING

This section details the loss function, network training, and optimization approaches of DRLB. Denote the encoder and decoder functions of the Pattern and Background Nets as $f_{\phi^P}, f_{\theta^P}, f_{\phi^B}, f_{\theta^B}$, respectively. Kingma and Welling [2013] suggested that the MLE of $P(X)$, i.e. optimization of $Z^P$ and $Z^B$, could be alternatively achieved by maximizing the lower bound as derived in (6).

**Maximize the expectation term**. The first term in (6) is the expectation $\mathbf{E}_{Q_{\phi^P, \phi^B}(Z^P, Z^B|X)}[\log P_{\theta^P, \theta^B}(X|Z^P, Z^B)]$, which could be maximized by training the two networks. The posterior probability $P(Z^P, Z^B|X)$ could be approximated by $Q_{\phi^P, \phi^B}(Z^P, Z^B|X)$. For the log-likelihood term $\log P_{\theta^P, \theta^B}(X|Z^P, Z^B)$, noting $X$ is the sum of $X^P$ and $X^B$ and it only contains binary values, we formulate it using the following Bernoulli distribution:

$$P_{\theta^P, \theta^B}(X_{ij}|Z^P, Z^B) \sim Ber(f_{\theta^P}(Z^P)_{ij} + f_{\theta^B}(Z^B)_{ij})$$
(7)

Denote the sum of the outputs of the two decoders as $f_\theta(Z)_{ij} \triangleq f_{\theta^P}(Z^P)_{ij} + f_{\theta^B}(Z^B)_{ij}$. The log-likelihood term can be written as:

$$\log P_{\theta^P, \theta^B}(X|Z^P, Z^B) = \sum_{ij} X_{ij} \log(f_\theta(Z))_{ij}$$
$$+(1 - X_{ij}) \log(1 - f_\theta(Z))_{ij} \quad (8)$$

Maximizing the log-likelihood is equivalent to minimizing the binary cross-entropy loss. We define the loss function for the expectation term in (6) as:

$$L_{recon1} = -\frac{1}{mn} \log P_{\theta^P, \theta^B}(X|Z^P, Z^B) \quad (9)$$

**Optimize the KL divergence term**. In variational inference, the KL divergence terms in (6) regularize the discrepancies between the distributions of the latent representations $Z^P$ and $Z^B$ and their prior distributions [Kingma and Welling, 2013]. To enable an effective disentangled learning of $Z^P$ and $Z^B$, DRLB first approximates a prior distribution of $Z^B$ to ensure $f_{\theta^B}(Z^B)$ can accurately and specifically capture the background bias. A disentangled learning of $Z^P$ and $Z^B$ is achieved by maximizing the generation of $X$ via $f_{\theta^P}(Z^P) + f_{\theta^B}(Z^B)$. We set $P(Z^P) \sim N(0, I)$. For $Z^B$, DRLB approximates its prior distribution as detailed below.

**Estimate the distribution of bias and the prior distribution of $Z^B$**. To approximate the distribution of $Z^B$, we first generate a simulated bias matrix, denoted by $\hat{X}^B$. By (3), $P(X_{ij}^B = 1) \propto \mathbf{p}_i^c \cdot \mathbf{p}_j^r$, here $\mathbf{p}_j^r$ and $\mathbf{p}_i^c$ are two vectors that represent the row-wise and column-wise background probabilities. Noted, $\mathbf{p}^r$ and $\mathbf{p}^c$ are not identifiable when $p_{ij}^c$ are unknown. Thus, the probabilities are heuristically approximated by $\hat{\mathbf{p}}_i^c = \frac{X_{i:}}{n}$ and $\hat{\mathbf{p}}_j^r = \frac{X_{:j}}{m}$. $\hat{X}^B$ is further randomly generated by the product of the approximated probabilities with a hyper-parameter $\alpha$:

$$P(\hat{X}_{ij}^B = 1) = \alpha \cdot \frac{X_{i:}}{n} \cdot \frac{X_{:j}}{m} \quad (10)$$

As illustrated in Figure 1), the Background Net in DRLB learns the latent representation of the background bias, denoted as $\hat{Z}^B$, from $\hat{X}^B$. Similar to (8) and (9), the generative model can be trained using the following loss:

$$L_{recon2} = -\frac{1}{mn} \log P_{\theta^B}(\hat{X}^B|\hat{Z}^B) \quad (11)$$

, here $\hat{Z}^B = f_{\phi^B}(\hat{X}^B)$ is a prior distribution of $Z^B$ learned from the randomly simulated background bias matrix $\hat{X}^B$, and $\log P_{\theta^B}(\hat{X}^B|\hat{Z}^B) = \sum_{ij} \hat{X}_{ij}^B \log(f_{\theta^B}(\hat{Z}^B))_{ij} + (1 - X_{ij}) \log(1 - f_{\theta^B}(\hat{Z}^B))_{ij}$ is the log-likelihood function. To ensure a high robustness and flexibility of the method, we introduced the third loss based on Maximum Mean Discrepancy (MMD) to further regularize the discrepancies of $Z^B$ and $Z^P$ with their prior distributions [Gretton et al., 2012](see details in APPENDIX A.2):

$$L_{dist} = MMD^2(f_{\phi^P}(X), N(0, I))$$
$$+MMD^2(f_{\phi^B}(X), f_{\phi^B}(\hat{X}^B)) \quad (12)$$

With the above considerations, the dual networks in DRLB will be trained by minimizing the combined loss function:

$$L = L_{recon1} + L_{recon2} + \lambda L_{dist} \quad (13)$$

, where $\lambda$ is an adjustable hyper-parameter.

## 4 EXPERIMENTS

### 4.1 BENCHMARK AND BASELINES

We evaluate the performance of DRLB on both synthetic and real-world datasets. Noted, DRLB is a bias-removal method that could be seamlessly implemented with any BMF method. To show the effectiveness of DRLB in removing background bias and increasing the detection accuracy of BMF, we selected four BMF methods for the downstream BMF task, namely ASSO, PANDA, MEBF, and CG. As introduced in 2.1, ASSO [Miettinen et al., 2008] and PANDA [Lucchese et al., 2010b] are two classic methods that are commonly utilized as baselines in BMF method development. MEBF [Wan et al., 2020b] is a state-of-the-art (SOTA) method that has the top running speed and satisfactory accuracy. CG [Kovacs et al., 2020] is a SOTA method that robustly achieved the top prediction accuracy in multiple recent BMF works. Based on the experiments of recent BMF studies[Kovacs et al., 2020, Avellaneda and Villemaire, 2021], we consider the four selected BMF methods can represent conventional and SOTA methods. We want to note the goal of the experiment with these selected BMF methods is to demonstrate the effectiveness of bias-removal of DRLB, and the advantage of implementing DRLB with the BMF methods over applying BMF alone, rather than comparing DRLB against any of the BMF methods.

To show DRLB's superiority in background bias removal, we select BIND as the baseline method for comparison. To the best of our knowledge, BIND is the only baseline method for this type of analysis. For each data and BMF method, the performance is evaluated based on three different inputs: 1) the original input, 2) data with bias removed using BIND, and 3) data with bias removed using DRLB.

### 4.2 IMPLEMENTATION DETAILS

All the BMF methods and BIND used in this paper were implemented with the source code and default parameters from the original works. DRLB was implemented via PyTorch and trained with Adam Optimizer[Kingma and Ba, 2014]. In DRLB, five layers were set in each of the two neural networks, with dimensions $\{D, 200, 20, 200, D\}$ (here $D$ is the dimension of the input data). ReLU activation is used before and after the bottleneck layer. The final layer is mapped to binary values using a Sigmoid activation. Implementation details including batch size, learning rate, parameters, and hardware information are given in APPENDIX B. Analysis of running time and scalability is given in APPENDIX C.

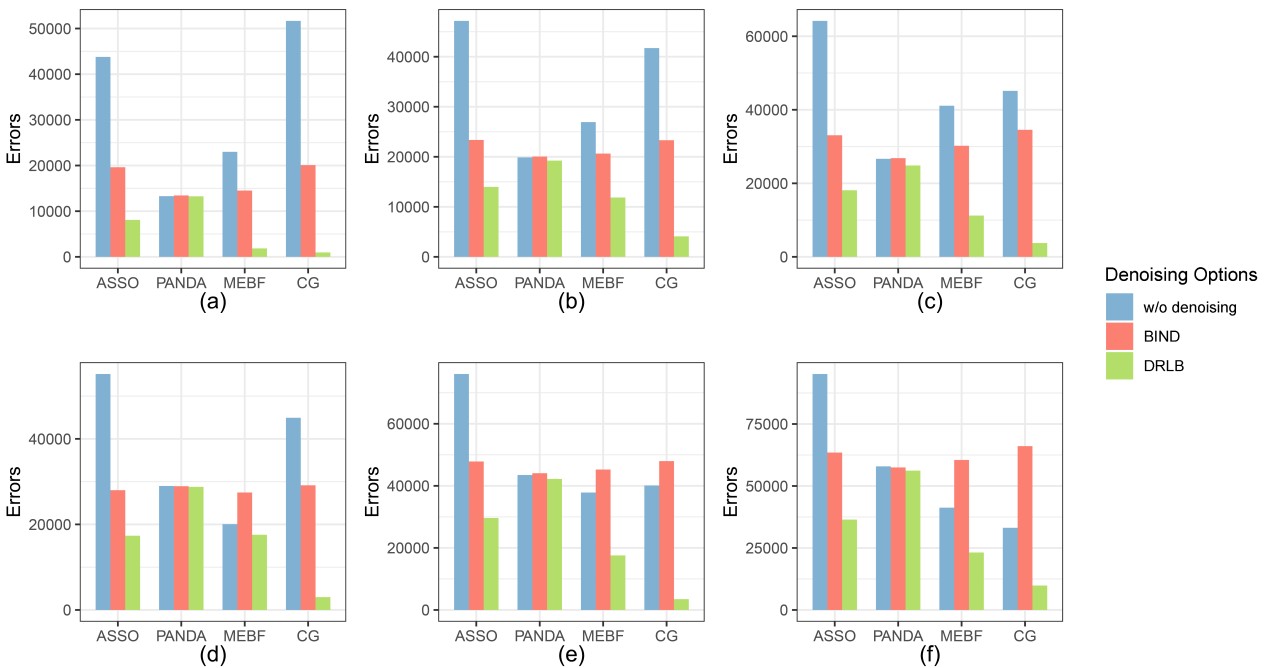

Figure 2: Comparison of reconstruction errors on simulated data. First row: pattern size = 80. Second row: pattern size = 120. Columns from left to right correspond to the number of simulated patterns = 2, 3, and 4. The three denoising settings, namely without denoising, BIND, and DRLB, are blue, red, and green colored, respectively.

## 4.3 EXPERIMENTS ON SIMULATED DATA

### 4.3.1 Experimental Settings.

We simulated Boolean matrices $X^{500\times500}$ of 500 rows and columns by $X = U \otimes V + X^B + E$, with pattern size $\in \{80, 120\}$, pattern number $k \in \{2, 3, 4\}$ and a fixed flipping error $E\colon\ = p(1 \rightarrow 0) = p(0 \rightarrow 1) = 0.01$. The row- and column-wise probability vectors of background bias were simulated by random sampling from the uniform distribution $\mathbf{p}^r, \mathbf{p}^c \sim U[0.1, 1]$. In total, 600 synthetic input matrices of six scenarios were simulated.

Two metrics were used in our evaluation. First, we evaluated the reconstruction error, which is the most common metric for evaluating the performance of BMF methods. The reconstruction error is defined as:

$$Reconstruction\ Error = ||U \otimes V - A^* \otimes B^*|| \quad (14)$$

, where $U, V$ generates the ground truth pattern matrix $X^P$ as defined in (3) and $A^*, B^*$ are decomposed patterns of a BMF algorithm. Second, we used the **signal**(1's in $U \otimes V$)/**noise**(1s from background and errors) ratio of the debiased matrix (bias-removed matrix) to evaluate the effectiveness of the bias removal of BIND and DRLB.

### 4.3.2 Evaluation on reconstruction error.

*DRLB drastically decreases the reconstruction error of BMF methods.* Figure 2 shows the reconstruction errors of different BMF methods on simulated data using different bias

removal approaches. We have seen that the performances of ASSO, MEBF, and CG are drastically improved when implemented on the DRLB-debiased data compared to the original input and BIND-debiased data. Also, a slight improvement was seen in PANDA. We note that DRLB is especially helpful for methods that have a high sensitivity in detecting dense patterns, such as ASSO and CG. These methods cannot distinguish between true patterns and dense blocks formed by background biases. Thus, they tend to suffer a higher false positive rate when there is a stronger background bias, which could be effectively handled by DRLB. On the contrary, BIND also showed a lower level of improvement compared to DRLB. In addition, it tends to lose too many pattern signals and may introduce additional biases Figure 3(b,e).

Our experiments showed that DRLB + CG achieved the lowest reconstruction error under all the testing scenarios. As shown in Figure 2, the reconstruction error of CG was high when applied to the original data, which drastically decreased when DRLB was implemented. CG is the top-performing method when applied to the data without bias, and it gains a drastically increased performance against bias when implemented with DRLB.

### 4.3.3 Evaluation on signal/noise ratio.

*DRLB can remove the majority of the background biases and achieves a high signal/noise ratio.* Table 1 shows the

| size | 80 | | | 120 | | |
|---|---|---|---|---|---|---|
| number | 2 | 3 | 4 | 2 | 3 | 4 |
| original | 0.18 | 0.29 | 0.37 | 0.43 | 0.68 | 1.02 |
| BIND | 0.06 | 0.10 | 0.05 | 0.48 | 0.15 | 0.53 |
| DRLB | **10.82** | **7.64** | **21.25** | **8.54** | **93.62** | **87.12** |

Table 1: Signal/noise ratio of the original input, BIND-debiased data, and DRLB-debiased data on simulated data with different pattern sizes and numbers of patterns.

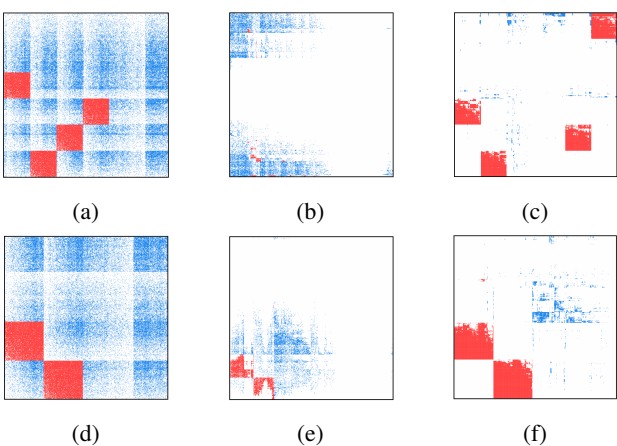

Figure 3: Bias-removing on simulated data. 1's in true patterns and background bias were red and blue colored, respectively. Row 1: four patterns of size 80. Row 2: two patterns of size 120. (a,d) original inputs ; (b,e) Bind-debiased data; (c,f) DRLB-debiased data.

signal/noise ratio of the original data, BIND-debiased data, and DRLB-debiased data in the simulated scenarios. It is noteworthy that the matrices debiased by DRLB have drastically increased signal/noise ratios compared to the original input or BIND debiased data. To illustrate the bias-removing power of DRLB, we visualized DRLB-debiased matrices versus the original inputs and BIND-debiased results of two simulation scenarios in Figure 3. It shows that the background bias was significantly removed by DRLB and the patterns become more distinct than the original and BIND-debiased matrices. BIND may remove entire rows or columns that are likely to be biased. Although BIND removes biases, it also loses a significant part of pattern signals, which explains why the signal/noise ratios of BIND-debiased matrices are even lower than the original input.

#### 4.3.4 Evaluation on estimating background bias.

We also evaluated the accuracy of DRLB in approximating the background bias. We first reconstructed the background bias matrix by the Boolean difference between the input matrix and the pattern matrices detected by BMF methods. Row- and column-wise bias levels were estimated by the frequency of 1s in each row and column of the reconstructed background bias matrix. We computed the correlation be-

tween the estimated bias level and the true bias level to evaluate the bias recovery capability of DRLB. Figure 4 shows the correlations of row- and column-wise bias on three simulated scenarios. The correlation between the estimated bias level and the true bias level is over 0.95 in most cases. The high correlation demonstrates that the row- and column-wise bias could be well captured by DRLB.

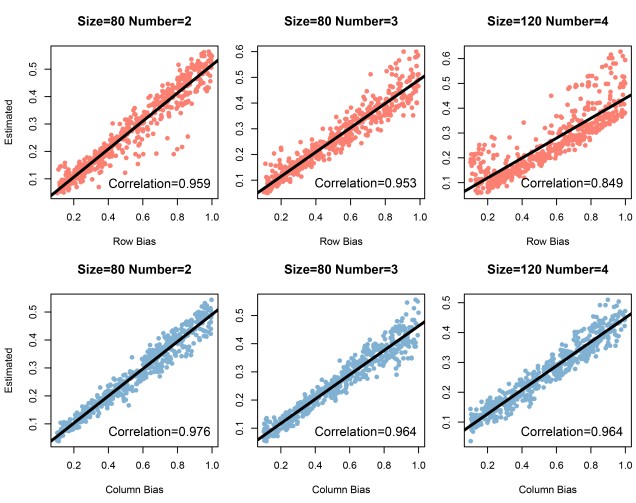

Figure 4: Correlation between estimated bias level and true bias level on three simulated datasets with various pattern sizes and numbers. First row: results of row-wise bias. Second row: results of column-wise bias on the same dataset.

### 4.4 ABLATION STUDIES

#### 4.4.1 Selection and sensitivity of hyper-parameters.

There are two hyper-parameters in DRLB, namely (1) $\lambda$ that balances contributions of the loss functions, and (2) $\alpha$ that controls the density of the generated background matrix. $\alpha$ depends on the data, which is recommended to be set between 2 and 3. In our experiments, we set $\alpha = 3$ for all the simulated data and $\alpha = 2$ for the real-world data. We also tested the impact of varied $\lambda$. Because DRLB + CG has been confirmed as the best-performed combination, we utilized CG for BMF in the test. Figure 5 shows the reconstruction errors of CG on the simulated scenarios when $\lambda$ varies from 0.1 to 1. We can see that our model is robust when $\lambda$ varies in a relatively wide range. But the reconstruction error tends to increase when $\lambda$ goes large. Therefore, we set the value of $\lambda = 0.3$ in all simulated and real-world data experiments.

#### 4.4.2 Influence by bias level.

In order to show the robustness of DRLB to different bias levels, we evaluated the method on simulated data with different bias levels. Four scenarios were simulated, with the row/column bias randomly sampled from $U[p, 1]$ and $p = 0, 0.1, 0.2, 0.3$. Still, DRLB + CG was utilized for the evaluation. Figure 6 shows the reconstruction error of CG

before/after denoising under these scenarios. Sampling from larger $p$ will increase the bias level and the reconstruction error. DRLB consistently performs well in handling the high-bias cases and ensures a small reconstruction error for the downstream BMF. We conclude that DRLB has a high robustness to varied bias levels. Therefore, it can be robustly applied to a wide range of binary data.

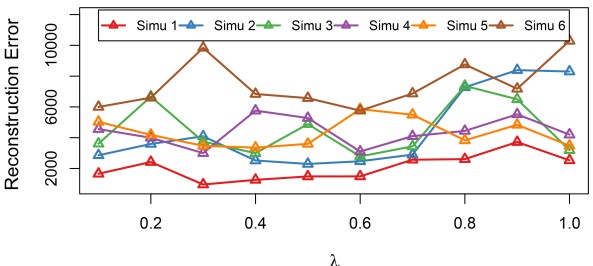

Figure 5: Reconstruction error of CG on six simulated scenarios after bias removed by DRLB using different $\lambda$. Different colors represent the six scenarios in Table 1.

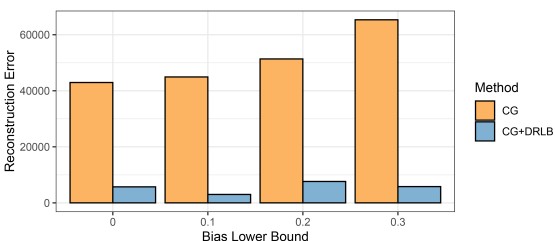

Figure 6: Performance of CG and DRLB + CG on simulated data of different bias levels.

### 4.5 EXPERIMENTS ON REAL WORLD DATA

To further evaluate the effectiveness of DRLB, we tested our model on two real-world single-cell RNA sequencing (scRNA-seq) data. BMF has been commonly applied in scRNA-seq data analysis [Chang et al., 2021, Fang et al., 2020]. Notably, scRNA-seq data (1) is non-negative, (2) contains a large number of 0s, and (3) always has heterogeneous row(gene)- and column(cell)-wise distributions (see detailed discussions in APPENDIX D), thus forming desired testing data for the BABMF problem. Both selected data were collected from liver cancer tissues[Ma et al., 2019b, Wang et al., 2019]. The two data have 17530 genes and 5762 cells (GSE125449) and 14452 genes and 4375 cells (GSE140228), respectively.

The low-rank patterns formulated by BMF correspond to functional modules in scRNA-seq data. In the real-world data-based experiment, we focus on demonstrating that the application of DRLB enables the detection of more biologically meaningful patterns. Still, DRLB + CG was selected for analysis and compared with CG.

#### 4.5.1 Data preprocessing.

For each data, we first select the top 2000 varied genes and cells based on row-/column-wise standard deviation, which gives us a $2000 \times 2000$ matrix. The original continuous data were binarized by setting the top 80% non-zero values to 1 and the other values to 0.

#### 4.5.2 Performance evaluation.

Since there is no ground truth for real-world data, instead of evaluating reconstruction errors and signal/noise ratio, we focused on demonstrating the biological meaning of the patterns derived from DRLB-debiased data vs original inputs. We first use the adjusted rand index (ARI) to test the coincidence between the detected patterns and known cell type labels. Intuitively, true patterns should have high ARI because most function modules are cell-type specific. On the other hand, false positives caused by background bias may not match well with cell types. Figure 7 shows the performance of DRLB + CG vs CG. Bias removal by DRLB resulted in a much higher ARI than applying CG merely to the original input. This result partially demonstrated that the implementation of DRLB improves the detection of functional gene modules.

To further evaluate the detected patterns, we performed a pathway enrichment analysis of the genes in each detected pattern against 3090 canonical pathways in the Molecular Signatures Database [Subramanian et al., 2005, Liberzon et al., 2011]. We found a large number of cancer-related pathways can only be detected by using the DRLB + CG. For example, in GSE125449, the PPAR signaling pathway was only detected in DRLB-debiased data. This pathway plays an important role in lipid circulation and metabolic reprogramming in cancer. Pathways related to immune response and other biological functions were also only detected by DRLB + CG. Our analysis demonstrates that the implementation of DRLB enables better bias handling and functional analysis of scRNA-seq.

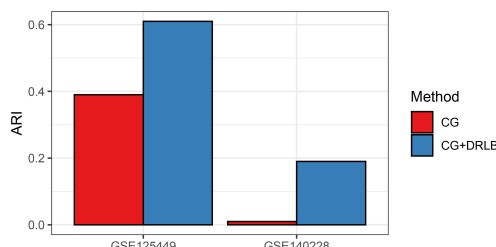

Figure 7: ARI on real-world data.

## 5 CONCLUSION

In this study, we introduce DRLB, a method that can effectively handle systematic biases in Boolean matrix factorization. By disentangling the input matrix into distinct pattern and background matrices, DRLB provides a more accurate representation of low-rank patterns in both simulated

and real-world data. DRLB can be seamlessly implemented with all existing BMF methods to improve their detection accuracy for biased data and enhance the reliability and context-specific meaningfulness of the detected low-rank patterns. The scope of this work is not to provide an efficient bias removal method for large data, and therefore one future task can be to provide a more efficient method of DRLB.

## Acknowledgements

This study is supported by National Science Foundation (DBI IIBR 2047631, IIS 2145314); and American Cancer Society (RSG-22-062-01-MM).

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

# Bias-aware Boolean Matrix Factorization Using Disentangled Representation Learning (Supplementary Material)

**Xiao Wang**[1]    **Jia Wang**[1]    **Tong Zhao**[3]    **Yijie Wang**[1]    **Nan Zhang**[4,5]    **Yong Zang**[2]    **Sha Cao**[2]    **Chi Zhang**[2]

[1]Department of Computer Science, Indiana University, Bloomington, Indiana, USA
[2]School of Medicine, Indiana University, Indianapolis, Indiana, USA
[3]Uber Inc, Seattle, Washington, USA
[4]Institute of Science and Technology for Brain-inspired Intelligence, Fudan University, Shanghai, China
[5]School of Data Science, Fudan University, Shanghai, China

## A  MATHEMATICAL DERIVATIONS

### A.1  THE DERIVATION OF EQUATION (6)

Here we show how we derived Eq.(6) from Eq.(5). With the disentanglement strategy, the latent space is decomposed into two independent components, $Z^P$ and $Z^B$. Because of the independence, we have: 1) $P(Z^P, Z^B) = P(Z^P)P(Z^B)$; 2) $Z^P$ and $Z^B$ are only associated with their own corresponding networks. Then:

$$
\begin{aligned}
\log P(X) &\geq \mathbf{E}_{Q_\phi(Z|X)}[\log P_\theta(X|Z)] - \mathbf{KL}(Q_\phi(Z|X)||P(Z)) \\
&= \mathbf{E}_{Q_{\phi^P,\phi^B}(Z^P,Z^B|X)}[\log P_{\theta^P,\theta^B}(X|Z^P,Z^B)] - \mathbf{KL}(Q_{\phi^P,\phi^B}(Z^P,Z^B|X)||P(Z^P,Z^B)) \\
&= \mathbf{E}_{Q_{\phi^P,\phi^B}(Z^P,Z^B|X)}[\log P_{\theta^P,\theta^B}(X|Z^P,Z^B)] - \mathbf{KL}(Q_{\phi^P}(Z^P|X)||P(Z^P)) - \mathbf{KL}(Q_{\phi^B}(Z^B|X)||P(Z^B))
\end{aligned}
\tag{15}
$$

### A.2  THE FORMULATION OF EQUATION (12)

In Eq.(12), we used Maximum Mean Discrepancy (MMD) to further constrain the distance of two distributions. Here MMD is a non-parametric metric that estimates discrepancies of different distributions by projecting data to a reproducing kernel Hilbert space with kernel functions. Concretely, let $X$ and $Y$ be two sets of samples with distribution $p$ and $q$, respectively. $x$ and $x'$ are different samples from $X$, $y$ and $y'$ are different samples from $Y$. The MMD between these two distributions is defined as:

$$
MMD^2(p,q) = \mathbf{E}_{x,x'}[k(x,x')] - 2\mathbf{E}_{x,y}[k(x,y)] + E_{y,y'}[k(y,y')]
\tag{16}
$$

where $k$ is some pre-defined kernel function.
In practice, the expectations can be estimated by sample means:

$$
MMD^2(X,Y) = \frac{1}{m^2}\sum_{i,j=1}^{m} k(x_i,x_j) - \frac{2}{mn}\sum_{i,j=1}^{m,n} k(x_i,y_j) + \frac{1}{n^2}\sum_{i,j=1}^{n} k(y_i,y_j)
\tag{17}
$$

where $m$ and $n$ are sample sizes of $X$ and $Y$.
In Eq.(12) the loss function $L_{dist}$ utilized MMD to constrain the distribution distances of two pairs of samples: 1) $f_{\phi^P}(X)$ and random samples from $N(0,I)$; 2) $f_{\phi^B}(X)$ and $f_{\phi^B}(\hat{X}^B)$.

## B  IMPLEMENTATION DETAILS

Following the implementation information in 4.2, we provide further implementation details including batch size, learning rate, hyper-parameters, and hardware information below.

To train the two networks in DRLB, we used a batch size of 8 for simulated data and a batch size of 32 to accelerate the training for real-world data. The initial learning rate is set as 0.001, with a decay rate of 0.5 every ten epochs. Based on the experiments, we found the model converges well, and training for 100 epochs is enough for all the tested data. The hyperparameter $\lambda$ is set as 0.3 for all the data, and $\alpha$ is set between 2 and 3. We performed 10 runs and reported the averaged results for each input data.

All analyses were conducted on a laptop with 12th Gen Intel(R) Core(TM) i7-12700 CPU and NVIDIA GeForce RTX 3060 Ti GPU.

## C    RUNNING SPEED AND COMPUTATIONAL EFFICIENCY

DRLB is a deep neural network-based method that relies on GPU computation. Noted, DRLB is designed to be implemented with BMF methods. Thus, instead of deriving the theoretical computational speed of the DRLB algorithm, we evaluated its running speed on our testing data and compared its running time with BMF methods. Among the SOTA BMF methods, MEBF is one of the fastest methods while ASSO, PANDA, and CG have relatively slower but acceptable running speeds. In our analysis, we have seen that DRLB is generally faster than or at the same level as the BMF methods ASSO, PANDA, and CG. On both simulated and real-world data, the running time of DRLB is about 10 times of MEBF. Considering MEBF is a highly scalable method, we concluded that the running of DRLB does not significantly introduce additional running costs when implemented with existing BMF methods and it can be applied to large data sets. Also, DRLB has a similar running speed compared with BIND.

## D    SYSTEMATIC BIASES IN SINGLE-CELL RNA-SEQUENCING DATA

Single-cell RNA-sequencing (scRNA-seq) data measures relative gene expression (or transcriptomic) abundance in a group of single cells. The typical form of single-cell RNA-sequencing (scRNA-seq) data is a matrix, in which each row is a gene feature and each column is a single cell. Each element in the data measures the relative expression level of a gene in a cell. Noting the high sparsity of scRNA-seq data, binarization is commonly utilized in scRNA-seq data processing and analysis. A common binarization approach is using a hard cutoff. All values larger than the cutoff are assigned as 1 and all the other values are assigned as 0.

Here we want to discuss the binarized scRNA-seq data generally contain systematic biases as described in Eq.(2) and Eq.(3), especially for the ones generated by using the 10x Chromium or other drop-seq protocols [Andrews et al., 2021, Jovic et al., 2022]. Specifically, 10x Chromium and other drop-seq protocols generate a pooled library of 5000-20000 single cells and then sequence the pooled library to measure a cell-wise gene expression profile. In the pooled library, some cells may have more mRNA molecules amplified and measured while some others may have a lower mRNA amplification rate and less mRNA measured, majorly because of the stochastics of biochemical reactions of amplification and sequencing. Thus, in the 10x Chromium and other drop-seq data, the total signal (or called total counts) collected from each cell (column) could be varied a lot. The variation of this total signal is similar to the total number of items purchased by different users in purchase history data. Some "super-cells" have much higher total signals (total counts) measured compared to other cells, which are like the super-buyers who tend to purchase more items in purchased history data. This variation will be inherited in the binarized data. Hence, the variation of the total measured signals through different cells naturally forms a column-wise bias in binarized scRNA-seq data, which follows Eq.(2) and Eq.(3). On the other hand, the exact mRNA abundance of some genes, such as metabolism, cell structure, and other housekeeping genes, are always high while some genes like transcriptional factors or signaling molecules always have low expression levels in cells. Similarly, this gene-wise variation will be inherited in the binarized data. Thus, the difference in the natural expression level of different genes forms a row-wise bias in binarized scRNA-seq data, which follows Eq.(2) and Eq.(3).

In summary, the scRNA-seq data generated by using 10x Chromium and other drop-seq protocols naturally contains bias led by the varied distribution of row-wise and column-wise signals. Thus, this data type formed a desired real-world testing data type to benchmark DRLB. In this study, both of the selected testing data sets, GSE125449 and GSE140228, were generated by using the 10x Chromium or drop-seq protocols.