# OpenReview forum: "Bias-aware Boolean Matrix Factorization Using Disentangled Representation Learning"
_auai.org/UAI/2024/Conference — UAI 2024 poster_

### Official Review · Reviewer_TQFk · 2024-03-15

**Q2-1 Originality-Novelty:** 3
**Q2-2 Correctness-Technical Quality:** 3
**Q2-5 Clarity Of Writing:** 3

**Q1 Summary And Contributions:**

This paper proposes a new approach to preprocess Boolean data in the context of Boolean matrix factorization (BMF). Although the proposed approach, dubbed DRLB, performs well on a series of experiments, the mathematical motivation is, in my opinion, rather weak. I would think that a more application-oriented venue would be more suitable.

--> However, authors have clarified the motation, and although there is no theory guaranteeing recovery (e.g., under some simplified model), I appreciate the contribution and have updated my score.

**Q2-3 Extent To Which Claims Are Supported By Evidence:**

3: Good: the main claims are supported by convincing evidence (in the form of adequate experimental evaluation, proofs, (pseudo-)code, references, assumptions).

**Q2-4 Reproducibility:**

4: Excellent: key resources (e.g. proofs, code, data) are available and key details (e.g. proof sketches, experimental setup) are comprehensively described for competent researchers to confidently and easily reproduce the main results.

**Q3 Main Strengths:**

The proposed approach to unbias Boolean data seems to perform well on a series of numerical experiments, performing better than the state of the art.

**Q4 Main Weakness:**

- It is rather unclear why using a neural network is needed for this problem. Why does it make sense to have a hidden variable Z? In fact, there is no theoretical motivation behind this choice. Does this approach bring any theoretical guarantee?
- Definition 1 (Bias-Aware BMF) is not rigorous: X is introduced as a Boolean matrix, but then it is assumed to be a random variable, P(X_{ij} = 1) is equal to some quantity. A similar problem arises around Equation 4.
Also p_{ij} is not well defined... just it should be positive when the corresponding entry of UoV is.
- It is not very clear how the DRLB approach works exactly, and what it outputs. Is it a Boolean matrix? (Since sigmoid is used at the output layer) Why choose such parameters for the neural network? How sensitive it is to the architecture? Why are 20 hidden neurons enough for all experiments? And 200 in the intermediate layers? Should we increase these numbers for larger data sets? There is no discussion about this.

--> these issues have been addressed by the authors.

**Q5 Detailed Comments To The Authors:**

Other comments:
- Product of two lower-rank Boolean matrices: since X = U * V (in the noiseless case), the rank of X is might actually be smaller than that of U and V. It would be more precise to say "Product of two smaller Boolean matrices".
- Authors should better explain and define what they mean by "disentangle" as this is not very clear, at least to me.
- Authors introduce the notation with upperscript letters, e.g., U^{m \times k}, to mean that U has dimension m times k. But then they use X^0, X^B, X^P as well. This is rather confusing. Later X^L is also introduced (after equation 14) but it is not defined.
- Equations are part of the text and should be properly punctuated.
- The notation U^{m \times k} o V^{k \times n}_{ij} is confusing (after equation 3).
- I would recommend the authors to make they code available online.

--> these issues have been addressed by the authors.

**Q9 Complying With Reviewing Instructions:**

Yes

---

> ### Author Rebuttal · Authors · 2024-04-03
>
> Response to reviewer TQFk's questions and concerns (Part I):
>
> We appreciate your time and effort in reviewing our manuscript. Your comments and questions provide helpful information for us to improve the presentation of our work. We appreciate that you identified that our method achieved improved performance compared to baselines. However, based on the weakness listed by you, we think that you may have a significant misunderstanding of this work. All of your major questions are about the motivation and rationale of our problem formulation and method setup. All minor questions are about presentations. We believe these questions could be smoothly solved in this rebuttal phase by providing more explanations.
>
>
> First, we hope to ask for a kind agreement that the differences in notations and presentations of the same problem caused by the different presentation habits among fields (like statistics, computer science, and even computational biology) could be tolerated, especially when a computational task is clearly defined.
>
>
> BMF is a widely used first-line method for Boolean/Binary data analysis. We want to note this work aims to tackle an under-studied topic in BMF, namely awareness of bias caused by imbalanced row-/column-wise distributions, which may affect the robustness and accuracy of classic BMF methods.
>
>
> In this manuscript, we first provided a review and rigorously defined formulation of the problem. We discussed the limitation of classic probabilistic formulation in handling this problem. Then, we provided a solution from a novel perspective of variational inference and disentangled learning. We provided rigorous formulation of the problem and derivation of the solution in 3.3 and 3.4, i.e., (1) how a Boolean matrix generated by the sum of bias and true signals could be formulated under variational inference, and (2) how signals from bias could be detected and removed from the input matrix by disentangled representation learning.
>
>
> Please see our detailed point-to-point response to your questions below:
>
>
> Main concern 1: It is rather unclear why using a neural network is needed for this problem. Why does it make sense to have a hidden variable Z?  In fact, there is no theoretical motivation behind this choice. Does this approach bring any theoretical guarantee?
>
>
> Responses:
> In 3.2, under formula (3), we discussed that because $p_{ij}$ can hardly be specified in a real-world problem, it is hard to find a solution via a classic probabilistic model. As mentioned in the manuscript, this work aimed to provide a novel solution from the perspective of variational inference and disentangled learning. Variational inference is well-studied and utilized in representational learning.
>
> The introduction of latent representation Z in formula (5-6) follows standard variational inference theories that have been commonly utilized in variational autoencoder (VAE). See details in the reference of Kingma and Welling 2013. It provides a general generation approach of X in this study. We further adopted the idea of disentangled representational learning, as supported by the related works provided in 2.3, to separate the generation of the pattern and bias matrices, i.e. $Z^P$ and $Z^B$ (formula (6) and (7)).
>
> The utilization of neural networks is a common approach in variational inference to solve the latent representation, i.e. solving $Z^P$ and $Z^B$ in this problem.
>
> We want to note the theoretical motivation and mathematical rigorousness of our method are rigorously and well supported by the theories of variational inference and our derivations in 3.2-3.4.
>
>
> Main concern 2: Definition 1 (Bias-Aware BMF) is not rigorous: X is introduced as a Boolean matrix, but then it is assumed to be a random variable, $P(X_{ij} = 1)$ is equal to some quantity. A similar problem arises around Equation 4. Also $p_{ij}$ is not well defined... just it should be positive when the corresponding entry of UoV is.
>
>
> Response:
> We respectfully disagree with your concern about the rigorousness of Definition 1. In statistics, observed data, such as a Boolean vector or matrix, is considered an observation of a random variable. The "given Boolean matrix X" under definition 1 for sure means observed data of a random variable, especially when we provided the generation approach of X in the formula (2) and mentioned, "we further extend (2) to the following probabilistic definition of the bias-aware BMF problem".
>
> We have discussed the uncertainty of $p_{ij}$ leads to the main challenge of the task. We mentioned the problem formulated by formula (3) could be solved by MLE if $p_{ij}$ can be defined. However, because $p_{ij}$ cannot be defined in most real-world cases, it motivated us to develop this method to solve the problem from a different perspective.

---

### Official Review · Reviewer_wv8L · 2024-03-20

**Q2-1 Originality-Novelty:** 2
**Q2-2 Correctness-Technical Quality:** 3
**Q2-5 Clarity Of Writing:** 3

**Q1 Summary And Contributions:**

This paper proposes a disentangled representation learning method for binary matrix factorization that considers bias, or data imbalance, and shows the usefulness with several experiments. The idea of bias-aware binary matrix factorization sounds like interesting, and the experiments are conducted appropriately to support the proposed method.

**Q2-3 Extent To Which Claims Are Supported By Evidence:**

3: Good: the main claims are supported by convincing evidence (in the form of adequate experimental evaluation, proofs, (pseudo-)code, references, assumptions).

**Q2-4 Reproducibility:**

2: Fair: key resources (e.g. proofs, code, data) are unavailable but key details (e.g. proof sketches, experimental setup) are sufficiently well-described for an expert to confidently reproduce the main results.

**Q3 Main Strengths:**

- The idea of disentangled learning for bias-aware binary matrix factorization sounds like appropriate to improve the conventional binary matrix factorization.

- The experiments are well designed and thorough to support the proposed method.

**Q4 Main Weakness:**

- The argument should be a little bit refined as humble as possible. The bias that this paper deals with is not related with fairness, and in some sense, it's just a sort of imbalanceness of the data distribution.

- The proposed method needs some theoretical justification on the proposed architecture.

**Q5 Detailed Comments To The Authors:**

- The proposed dual encoder-decoder architecture needs to be justified to work out the bias appropriately. Even though it might work in some level of imbalance of data distribution, there is nothing on the theoretical justification.

- The definition of bias should be clearly made in the problem formulation.

- Some experiments on the disentangled representation might be helpful to support the proposal.

**Q9 Complying With Reviewing Instructions:**

Yes

---

> ### Author Rebuttal · Authors · 2024-04-05
>
> We appreciate your time and effort in reviewing our manuscript. We appreciate that you identified our work's significance, novelty, and performance. We also appreciate your careful reading of the manuscript. Your comments are very helpful in improving the quality of our manuscript. After carefully reviewing your comments and suggestions, we identified your comments mainly related to the theoretical justification of the proposed disentangled learning and related experiments if no theoretical proof can be provided. Please see our response below:
>
> Main concern 1: The argument should be a little bit refined as humble as possible. The bias that this paper deals with is not related with fairness, and in some sense, it's just a sort of imbalanceness of the data distribution.
>
> Response: Thanks for pointing this out. We agree with you. The word "fair" is only mentioned one time in the conclusion. We will remove this word in the revised version. Our original thinking is that the patterns identified by conventional methods from the bias-containing data are less fair because they tend to identify the rows or columns of more 1s (like the super items or super users in purchase history data).
>
> Main concern 2: The proposed method needs some theoretical justification on the proposed architecture. And two extensions in Minor concerns: Minor concern 1 "The proposed dual encoder-decoder architecture needs to be justified to work out the bias appropriately. Even though it might work in some level of imbalance of data distribution, there is nothing on the theoretical justification. "; Minor Concern 3: "Some experiments on the disentangled representation might be helpful to support the proposal."
>
> Response: We appreciate these comments. We agree that the theoretical foundation of DRLB is not complete. However, we can discuss the theoretic support to a certain level. The theoretical foundation of DRLB includes two parts: (i) if the mathematical consideration of the generation approach of X under variational inference is valid and (ii) if the problem is always identifiable, i.e. if the DRLB approach can identify the patterns and bias. For i, we introduced formulas (5) and (6) in section 3.3 by following the standard variational inference theories which are commonly used in variational auto-encoders (VAEs). The mathematical discussion here justifies i) maximizing the likelihood by optimizing the three loss terms in our model; ii) signals in the observed data can be separated into two latent spaces by two different encoders. However, to enable the disentanglement of patterns and bias, there must be some constraints on the latent spaces. Thus, we consider that the mathematical consideration of the generation approach of X under variational inference is valid. But to ensure the problem is identifiable, additional assumptions must be introduced.
>
> By optimizing the expectation term, we have constrained the summation of the two network outputs to reconstruct the input. Referring to BIND, we could get an approximation of the bias distribution by utilizing the generation probabilities in formula (10), which helps to enable $Z^B$ to approximate the latent distribution of bias by using the MMD loss. Based on the combination of these two designs, the latent variable $Z^P$ must capture the latent distribution of the patterns.
>
> We admit that the solution part of DRLB is less rigorously supported. To theoretically ensure the identifiability of DRLB, one necessary condition is that bias distribution should be unbiasedly estimated. This part cannot be proved, or actually, the bias distribution cannot be unbiasedly estimated. So far we do not have a good mathematical derivation to support the bound of the difference between the estimated bias distribution and the true distribution (actually, this shares the same problem of the unknown $p_{ij}$ as mentioned in 3.2).
>
> We agree with the reviewer that more experiments on disentangled learning might be helpful to support the proposed method. We are conducting experiments on the disentangled learning part, aiming to validate (1) how pattern size, the density of 1s, and bias level may affect the disentanglement of pattern and bias, and (2) how biased estimation of the bias distribution may affect the disentanglement part. We will add the results to the final version.
>
> Minor concern 2: The definition of bias should be clearly made in the problem formulation.
> We think the bias has been rigorously defined in 3.2 Formula 2 and Definition 1. We will add more clarification that the bias considered here is caused by row-/column-wise varied signal levels.

---

### Official Review · Reviewer_J7n1 · 2024-03-27

**Q2-1 Originality-Novelty:** 2
**Q2-2 Correctness-Technical Quality:** 3
**Q2-5 Clarity Of Writing:** 3

**Q1 Summary And Contributions:**

This paper introduces Disentangled Representation Learning for Binary matrices (DRLB), a novel approach to Boolean Matrix Factorization (BMF) that  enhances the accuracy and interpretability of pattern extraction from binary data. DRLB leverages a dual auto-encoder network to disentangle true patterns from row- and column-wise biases. The method demonstrates superior performance in precision improvement over traditional BMF techniques across both synthetic and real-world datasets. This work not only addresses the critical issue of inherent biases in BMF but also opens new avenues for bias-aware data analysis, marking a significant advancement in the field.

**Q2-3 Extent To Which Claims Are Supported By Evidence:**

3: Good: the main claims are supported by convincing evidence (in the form of adequate experimental evaluation, proofs, (pseudo-)code, references, assumptions).

**Q2-4 Reproducibility:**

4: Excellent: key resources (e.g. proofs, code, data) are available and key details (e.g. proof sketches, experimental setup) are comprehensively described for competent researchers to confidently and easily reproduce the main results.

**Q3 Main Strengths:**

Innovative Approach to BMF: By introducing DRLB, the paper presents a novel methodology that  advances the field of Boolean Matrix Factorization (BMF).

Bias Detection and Correction: DRLB  identifies and corrects for inherent biases in binary datasets, addressing a significant challenge in traditional BMF methods that often overlook row- and column-wise signal distributions.

Deep Learning Integration: The integration of a dual auto-encoder network into the BMF process represents a significant strength, leveraging deep learning techniques to disentangle and decode binary matrices in a way that traditional methods cannot.

**Q4 Main Weakness:**

Computational Demands and Efficiency: The adoption of a dual auto-encoder network, though innovative, introduces significant computational demands. The paper does not sufficiently address these concerns, particularly regarding computational efficiency, scalability to very large datasets.

Lack of Comparative Analysis with Other Bias-Aware Methods: The study primarily compares DRLB with traditional BMF methods and briefly with BIND for bias correction. However, it does not provide a comprehensive comparative analysis against other modern bias-aware techniques in machine learning and data mining.

Exploration of Alternative Deep Learning Architectures: The choice of dual auto-encoders is central to DRLB's methodology, but the paper does not explore or discuss the potential of alternative deep learning architectures that might offer similar or improved performance. Such an exploration could uncover more efficient or effective solutions for disentangling patterns from binary data.

**Q5 Detailed Comments To The Authors:**

Some sections of the paper, particularly those describing the mathematical  and algorithmic processes of DRLB, could benefit from clearer explanations or step-by-step breakdowns. The choices behind specific methodological aspects, such as the architecture of the dual auto-encoders and the selection of loss functions, could be further justified with a discussion on their advantages over other potential choices. Expanding the comparative analysis to include a wider range of bias-aware methods and alternative deep learning architectures could provide a more comprehensive view of where DRLB stands in the landscape of BMF and related fields.

**Q9 Complying With Reviewing Instructions:**

Yes

---

> ### Author Rebuttal · Authors · 2024-04-06
>
> We appreciate your time and effort in reviewing our manuscript. We appreciate that you identified our work's significance, novelty, and performance, especially your recognition of our integration of disentangled learning in solving this task. We also appreciate your careful reading of the manuscript. Your comments are very helpful in improving the quality of our manuscript. After carefully reviewing your comments and suggestions, we identified your comments are mainly about further discussions or improvement of the manuscripts. Some questions are not easy to be addressed in a short rebuttal period. Here we provided some feedback. Again, we appreciate your constructive questions. Please see our response below:
>
> Major Concern 1: Computational Demands and Efficiency: The adoption of a dual auto-encoder network, though innovative, introduces significant computational demands. The paper does not sufficiently address these concerns, particularly regarding computational efficiency, scalability to very large datasets.
>
> Response: BMF methods are all heuristic because the problem is NP-hard. Among existing methods, MEBF is must faster compared to others in terms of running speed. MEBF increases running efficiency by sacrificing robustness. While our method, DRLB is slower than MEBF, it is at a similar level as other conventional methods. We admit the scope of this work is not to provide an efficient bias removal method for large data. Instead, considering the bias problem has not been well solved, this work aims to solve the problem by providing an accurate bias-removal method with an acceptable running speed. One future task can be to provide a more efficient method of this task. We will include this discussion in our revised manuscript.
>
> Major Concern 2: Lack of Comparative Analysis with Other Bias-Aware Methods: The study primarily compares DRLB with traditional BMF methods and briefly with BIND for bias correction. However, it does not provide a comprehensive comparative analysis against other modern bias-aware techniques in machine learning and data mining.
>
> Response: This is a very good point. Thanks for pointing this out. On one hand, the matrix data is unstructured while imaging and literature data are structured. Thus, we cannot find a good solution to our problem by adopting existing bias-aware of imaging and literature data in this short rebuttal period. However, on the other hand, we agree it will be good to compare the problem with the other bias-aware learning tasks. We will add this part to the related work in the revised manuscript.
>
> Major Concern 3: Exploration of Alternative Deep Learning Architectures: The choice of dual auto-encoders is central to DRLB's methodology, but the paper does not explore or discuss the potential of alternative deep learning architectures that might offer similar or improved performance. Such an exploration could uncover more efficient or effective solutions for disentangling patterns from binary data.
>
> Response:  Because the signal and bias are additive in the probabilistic model of the Bias-Aware BMF (definition 1), we consider the duel auto-encoders to be a straightforward model of the generation of observed Boolean/binary data. We agree with the reviewer that there could be alternative deep learning architectures. But we consider the duel network is one of the simplest models. We will add more discussions in the revised manuscript.
>
> In addition, a similar response was made to answer reviewer TQFk's question:
> (1) The setup of parameters and architecture of the neural network, including the number of neurons in each hidden layer, are mainly based on empirical experience. We want to respectfully remind the reviewer that almost all deep learning-based models may not provide how such parameters are selected. The settings of parameters based on empirical experiences can ensure performance under most circumstances. (2) Based on both past literature and our experiments, setting a different dimension of the bottleneck layer may not significantly impact the performance. Based on our experiments (the maximum input dimension is 2000), we do not see the number of neurons need to be increased unless the original dimension of the data is super large. (3) We have included a comprehensive ablation experiment of the important parameters in our model in section 4.4 to show the robustness of the model.
>
> Additional comments:
> Some sections of the paper, particularly those describing the mathematical and algorithmic processes of DRLB, could benefit from clearer explanations or step-by-step breakdowns. The choices behind specific methodological aspects, such as the architecture of the dual auto-encoders and the selection of loss functions, could be further justified with a discussion on their advantages over other potential choices.
>
> Response: We appreciate the reviewer's suggestion. We will revise the format of the equations to clearer versions. We will also add the discussions.

---

### Official Review · Reviewer_nqbt · 2024-03-29

**Q2-1 Originality-Novelty:** 2
**Q2-2 Correctness-Technical Quality:** 3
**Q2-5 Clarity Of Writing:** 3

**Q10 Ethical Concerns:**

None.

**Q1 Summary And Contributions:**

This paper treats binary matrices as the sum of low rank patterning, background bias, and heteroskedastic noise, where the bias may be row-wise or column-wise. The authors present Disentangled Representation Learning for Binary matrices (DRLB) as a denoising method to be applied before binary matrix factorization.  This novel method proceeds by running two neural networks (Pattern Net and Background Net) that are collaboratively trained on the input matrix and return two matrices that represent the pattern + errors and background bias, respectively.  The loss function represents the combined loss associated with a lower bound on the loglikelihood, the prior distributions for the pattern and background bias, and the discrepancy between the (pattern/background) matrices and their respective priors. The method is demonstrated on simulated data in conjunction with four SOTA BMF methods and compared to a recently proposed denoising method.  Their real-world application involved analysis of scRNA-seq data obtained from liver cancer tissue to identify low-rank patterns corresponding to functional modules that are biologically meaningful. Theoretical results are worked out and provided in the supplementary material.  I did not check all results thoroughly, but what I did check seems to be sound.

**Q2-3 Extent To Which Claims Are Supported By Evidence:**

3: Good: the main claims are supported by convincing evidence (in the form of adequate experimental evaluation, proofs, (pseudo-)code, references, assumptions).

**Q2-4 Reproducibility:**

2: Fair: key resources (e.g. proofs, code, data) are unavailable but key details (e.g. proof sketches, experimental setup) are sufficiently well-described for an expert to confidently reproduce the main results.

**Q3 Main Strengths:**

The novelty of this work is in (i) using two autoencoders to separate the latent representation of the observed binary matrix into pattern and background bias, and (ii) using maximum mean discrepancy to further discriminate between the pattern and bias distributions. Running time is comparable with competing denoising method BIND, but DRLB seems to vastly enhance the performance of SOTA binary matrix factorization methods, such as CG. Results are also interpretable.

**Q4 Main Weakness:**

Novelty is somewhat reduced since (i) decomposition of binary signal into pattern and background was first proposed in BIND; and (ii) disentangled learning was adapted from computer vision to the binary setting.

The simulation design seems to assume columns and rows are sorted in a particular fashion that the "pattern" appears in blocks.  I can understand this for image data (that pattern and background are spatially located together), but it is not clear why that would be the case for binary data, such as purchase history or scRNA-seq data.

Results in simulation and on scRNA-seq data focussed around understanding the pattern part of the data.  Did the authors look at the background bias data to see if it was interpretable?  To what extent is the pattern being classified as background?

**Q5 Detailed Comments To The Authors:**

The paper is generally written well so I do not have any minor editorial comments.

Do columns and rows need to be sorted by a relevant variable for DRLB to detect the patterns?  If the rows and/or columns in your simulations had been shuffled, would DRLB still work as well?

In BIND, entire "irrelevant" rows and columns are eliminated, or zeroed out, whereas DRLB only removes partial columns or rows, though entire columns and rows may be deemed not patter as well.  Is there a connection between performing lasso regularization and elastic net regularization on the binary matrix?

You state in Appendix C that MEBF is the fastest BMF method, and that the running time of DRLB is about 10 times of MEBF. But then you also say that DRLB is generally faster than or at the same level as ASSO, PANDA, and CG, all of which are relatively slower methods compared to MEBF.  I find this confusing.  Is DLRB much faster than MEBF or comparable to ASSO, PANDA and CG with respect to running time?  IF it is sometimes fast but at other times slow, can you characterize when this happens? For example, is it related to signal-to-noise ratio, or is it related to how "blocked" the pattern is in the matrix? Something else?

**Q9 Complying With Reviewing Instructions:**

Yes

---

> ### Author Rebuttal · Authors · 2024-04-05
>
> We appreciate your time and effort in reviewing our manuscript. We appreciate that you identified the significance and novelty of our work, and the importance of the problem we solved here. We also appreciate your careful reading of the manuscript. Your comments are very helpful in improving the quality of our manuscript. After carefully reviewing your comments and suggestions, we identified your comments are very provocative. Please see our point-to-point response to your questions:
>
> Main concern 1: Novelty is somewhat reduced since (i) decomposition of binary signal into pattern and background was first proposed in BIND; and (ii) disentangled learning was adapted from computer vision to the binary setting.
>
> Response: We admit the novelty of this work is limited to a new solution to a previously identified question. However, we also want to strengthen (1) the bias problem affects the performance of classic BMF on real-world data a lot, (2) the bias-aware BMF problem has not been thoroughly solved, and (3) this work kind of provided a satisfactory solution of the problem, especially considering our method can be implemented with any general BMF method. Other than novelty, we appreciate the reviewer's recognition of the utility of this work.
>
> Main concern 2: The simulation design seems to assume columns and rows are sorted in a particular fashion that the "pattern" appears in blocks. I can understand this for image data (that pattern and background are spatially located together), but it is not clear why that would be the case for binary data, such as purchase history or scRNA-seq data.
>
> Response: Thanks for this question. The rationale of the submatrix-shaped patterns and row-/column-wise biases in the two data types could be explained with domain backgrounds. First, binary data is one way to represent purchase history and scRNA-seq data because (1) the two data are always sparse, (2) binary data mainly captures the presence and absence of signals, which captures the most distinct signals. In purchase history data, each row is an item and each column is a user, 1s/0s means an item has been bought by a user or not. Users purchase items with underlying but unobserved reasons, which result in co-purchased items. For example, people purchase basketball because they are (1) basketball players, (2) sports facility suppliers, (3) basketball fans, etc. Then the items co-purchased by people having the same reason may form 1s enriched sub-matrices in the binarized data. Meanwhile, as mentioned in the manuscript, super-buyers and super-items cause row-/column-wise biases. Similarly, in scRNA-seq data, each row is a gene and each column is a cell, 1s/0s means if a gene is expressed in a cell or not. Genes are regulated by transcriptional regulatory signals (TRS) in cells. In each cell, the gene expression is determined by the current unobserved TRS in the cell. The genes that are actively co-regulated by the same TRS will be co-present in cells having the TRS. Thus, the co-regulated genes in the cells having the same TRS may form a 1s enriched sub-matrix. Similarly, some genes are commonly expressed and some cells have more signals (counts) observed, which causes row-/column-wise biases.
>
> Main concern 3: Results in simulation and on scRNA-seq data focussed around understanding the pattern part of the data. Did the authors look at the background bias data to see if it was interpretable? To what extent is the pattern being classified as background?
>
> Response: This is a very good question. In our simulated data-based experiment, we found a significant consistency of the estimated bias and the true bias set in the experiment (Figure 4). In scRNA-seq data, we actually found the bias can be explained by commonly expressed genes (row-wise bias) and differences in total counts observed in each cell (column-wise bias). We will put the result in the revised version. We did not find patterns that were classified as background in our simulated data-based experiment (as can be partially shown by the signal/noise ratio). We can further increase the noise level or decrease the pattern size to see when the pattern and background cannot be disentangled.
>
> Minor concern 1: row/column order.
> Response: No, the method does not depend on the order of rows and columns. Like BMF, our method works on unstructured data.
>
> Minor concern 2: If we can link the difference in methods to L1/L2 loss?
> Response: We appreciate this thinking. But actually, the reason that BIND cannot identify a good way to differentiate noise and signal is the limitation of the probabilistic-driven method when $p_{ij}$ is unknown. So it used a hard cutoff.
>
> Minor concern 3: running time.
> Response: We apologize for the confusion caused here. We meant DRLB costs about 10 times the running time of MEBF (DRLB is about 10 times slower MEBF). MEBF is a super fast method that may have some robust issues. The running time of DRLB is at the same level as other BMF methods.

---

### Meta-Review · Area_Chair_i1GK · 2024-04-17

The author rebuttals have cleared up some misunderstandings. Following the rebuttal phase, the reviews of the paper are uniformly positive, with reviewers particularly positive about the quality of the proposed approach and experiment results. On the negative side, the novelty seems to be only moderate.